# Quo Vadis: Is Trajectory Forecasting the Key Towards Long-Term Multi-Object Tracking?

**Patrick Dendorfer**    **Vladimir Yugay**    **Aljoša Ošep**    **Laura Leal-Taixé**

Technical University of Munich

{patrick.dendorfer, vladimir.yugay, aljosa.osep, leal.taixe}@tum.de

## Abstract

Recent developments in monocular multi-object tracking have been very successful in tracking visible objects and bridging short occlusion gaps, mainly relying on data-driven appearance models. While significant advancements have been made in short-term tracking performance, bridging longer occlusion gaps remains elusive: state-of-the-art object trackers only bridge less than $10\%$ of occlusions longer than three seconds. We suggest that the missing key is reasoning about future trajectories over a longer time horizon. Intuitively, the longer the occlusion gap, the larger the search space for possible associations. In this paper, we show that even a small yet diverse set of trajectory predictions for moving agents will significantly reduce this search space and thus improve long-term tracking robustness. Our experiments suggest that the crucial components of our approach are reasoning in a bird's-eye view space and generating a small yet diverse set of forecasts while accounting for their localization uncertainty. This way, we can advance state-of-the-art trackers on the *MOTChallenge* dataset and significantly improve their long-term tracking performance. This paper's source code and experimental data are available at https://github.com/dendorferpatrick/QuoVadis.

## 1 Introduction

Multi-object tracking (MOT) is a long-standing research problem with a wide range of applications, including real-time dynamic situational awareness for robot navigation [21, 15, 77, 44, 62, 10], traffic monitoring [69], studying animal behavior [52] and monitoring biological phenomena [3].
State-of-the-art MOT methods [75, 8, 4, 82, 74, 65] combine regression [75, 4] and combinatorial optimization [8] in conjunction with identity re-identification (ReID) models [32, 59, 8, 75, 66, 4] to track objects in the image space. Such approaches have been very successful for tracking visible objects and bridging *short-term* occlusions. However, as can be seen in Figure 1b, *long-term tracking* remains an open challenge: state-of-the-art methods successfully bridge $50\%$ of occlusions within one second, falling below $10\%$ when the occlusion extends for more than 3 seconds. This is often not reflected in standard benchmarks [15, 21, 69, 77], as long-term occlusions are statistically rare.
In the past, combining ReID models with simple motion models has been immensely helpful [15] for short-term tracking. Nonetheless, as the occlusion time becomes longer, the set of possible associations grows exponentially with the increasing gap length [54]. This combinatorial complexity hinders the ability of visual-based ReID models to disambiguate between objects. Consequently, we believe that ReID models are insufficient to resolve long-term occlusions. However, continued efforts to develop stronger appearance models will remain an important research direction in vision-based MOT. Tracking moving pedestrians during occlusions is challenging, and simple linear motion models fail since human motion is complex and driven by non-observable factors such as goals, intent, or simply preferences. Therefore, we propose an alternative in this work: using long-term trajectory forecasting in order to prune down the combinatorial search space of feasible trajectory continuations. As the main contribution of this paper, we carefully study *what is needed* to leverage trajectory forecasting for multi-object tracking, as we have recently witnessed rapid progress in learning-based

36th Conference on Neural Information Processing Systems (NeurIPS 2022).

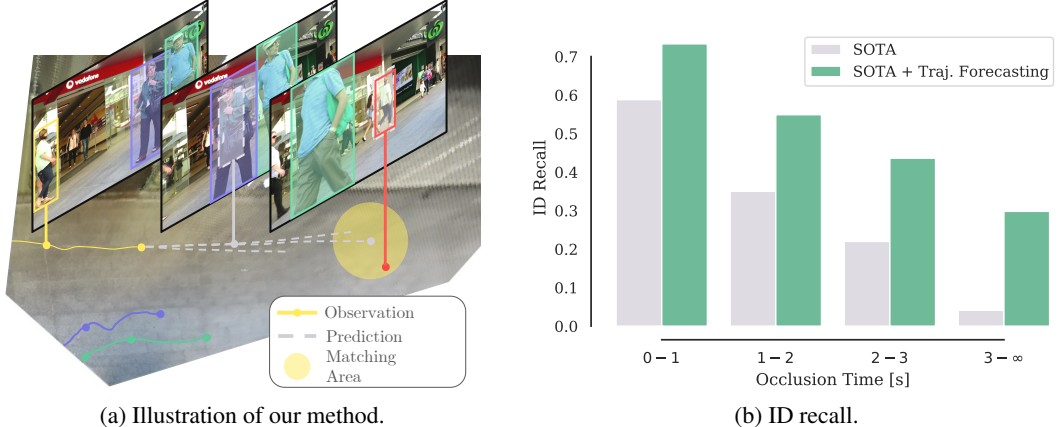

(a) Illustration of our method.      (b) ID recall.

Figure 1: State-of-the-art methods for vision-based MOT can successfully track visible objects and bridge short occlusion gaps; however, they fail at long-term tracking. (a) To bridge occlusion gaps, we lift monocular 2D detections to 3D world space, in which we reason about their possible future locations. This transformation allows us to reconnect detections that undergo long occlusions. As **yellow track** becomes occluded, our method predicts a small set of plausible future locations in 3D. In turn, we correctly associate **red detection** to the **yellow track** by accounting for the forecast uncertainty area. (b) As can be seen from the ratio of correct track association after different occlusion time lengths for the **prior work** and **our method**, this approach allows us to improve long-term tracking capabilities and gap longer occlusion gaps significantly. *Best seen in color.*

trajectory forecasting [55, 38, 23, 1]. However, these methods operate in a fully-observed, metric bird's-eye view (BEV) space, effectively disentangling the effect of the perspective projection on reasoning about motion. By contrast, monocular MOT methods only observe a projection of the visible portion of our 3D space. Our analysis reveals that we can bridge this gap by localizing trajectories in BEV-space, but crucially, the localization of 2D bounding boxes in BEV must be *temporally coherent*. We achieve this by estimating a single homography per sequence in a data-driven manner.

Forecasting methods can reason beyond simple linear extrapolations, predict multiple possible future outcomes, and account for social interactions. But are these all necessary ingredients for bridging complex and long-term occlusions? Our study suggests that the key ingredient is to estimate a set of forecasts that can possibly cover several diverging future paths with only a handful of samples and account for prediction uncertainty.

Our *trajectory forecasting* approach can be applied to improve the long-term tracking capabilities of existing object tracking methods. In particular, by applying our framework on top of the state-of-the-art method [80] of the *MOTChallenge* benchmark, we improve the performance on HOTA on *MOT17* by 0.09pp and *MOT20* by 0.10pp and further decrease the number of IDSW by 93 and 36, respectively. We hope our conclusions will encourage the community to continue investigating how 3D reconstruction and trajectory forecasting improve single-camera long-term tracking.

We summarize our **main contributions** as follows: we (i) present a study on how we can reconcile two related fields of research on vision-based trajectory forecasting and monocular multi-object tracking. Our study reveals that the core component of this interplay is temporally coherent reasoning about motion in 3D space. We (ii) utilize a synthetic MOT dataset to study how to localize objects in 3D BEV space in a manner that facilitates robust reasoning about plausible future motion and which are the core forecasting components needed to bridge longer occlusion gaps; Finally, (iii) we demonstrate that we can generalize our conclusions from synthetic sandbox to real-world monocular *MOTChallenge* sequences and demonstrate that our recipe can be used to improve long-term tracking performance for several object trackers.

## 2 Preliminaries

This section discusses the fundamentals of vision-based multi-object tracking and trajectory forecasting, the current state-of-the-art, and analyzes failure cases.

## 2.1 Multi-object Tracking

Monocular multi-object tracking (MOT) is the task of localizing objects as bounding boxes in image sequences and assigning them an identity-preserving unique ID. State-of-the-art methods decompose the problem into object detection and detection association.

**Quantifying tracking errors.** The *detection* aspect of the task is commonly quantified by counting per-frame detection errors over the sequence. To quantify *association* errors, we count *identity switches* (IDSW) (*i.e.*, wrong ID swaps or re-initializing a ground-truth track with a different tracking ID) and *identity transfers* (IDTR) (*i.e.*, incorrectly linking two different objects with the same tracking ID). While a successful association over occlusion gaps decreases the number of IDSW, a wrong association between tracklets leads to an IDTR instead. Recently introduced HOTA [41] metric separately evaluates object detection and temporal association aspects of the tracking task. Temporal association is quantified via *association accuracy* (AssA) term, that quantifies *association recall* (AssRe) and *association precision* (AssPr). The AssRe term accounts for IDSW errors, while AssPr accounts for IDTR errors.

**Are all identity errors created equal?** Correct track association of objects that undergo longer occlusion gaps is especially challenging because the appearance and position of an object may drastically change. In MOT datasets [15, 21] the majority of occlusions are short occlusions ($\leq 2s$). Hence, solving rare long occlusions ($> 2s$) does not significantly impact the model performance. As a result, long-term tracking is commonly overlooked in the literature. This can be seen in Figure 1b: state-of-the-art methods bridge less than $10\%$ gaps beyond three-second-long occlusions.

**Prior work.** Early tracking methods focus on combinatorial optimization [79, 27, 36, 71, 49] and hand-crafting visual and motion-based descriptors [48, 33, 11], especially beneficial in the era of unreliable object detectors. State-of-the-art methods for monocular visual MOT are data-driven and primarily rely on appearance. Regression-based methods [4, 75, 82] can localize objects even when object detections are missing, often used in conjunction with ReID models to bridge short occlusions. However, regression models fail when an occluded person appears at a distant image position. For solving long-term occlusion, discrete optimization methods, combined with end-to-end learning based on graph neural networks [8, 70, 78], construct large graphs stretching over multiple seconds leading to high computational costs and complexity. Motion has always played an essential role in visual tracking [5, 20, 4], especially beneficial in 3D where it is dis-entangled from projective distortion [34, 50, 26]. The interplay between reasoning in 3D for monocular pedestrian tracking and linear motion models was first investigated in [29].

Identity preservation is important in several applications, ranging from video editing, safety camera analysis, and social robots interacting with humans to autonomous driving. We only have access to a single RGB camera in several application scenarios. Exceptions are autonomous driving datasets [21, 10] that generally provide 3D sensory data, together with 3D track information. However, only a handful of object tracks contain occlusion gaps longer than $2s$: $0.6\%$ in BDD100K [77] and $4\%$ in widely-used KITTI tracking [21] dataset. Therefore, autonomous driving datasets are, at the moment, not well suited for studying long-term tracking. Instead we conduct our experiments and analysis using *MOTChallenge* [15] dataset, where $19.4\%$ of tracks undergo *long* ($> 2s$) occlusions gaps.

We hypothesize that bridging long-term gaps requires understanding the projection geometry and motion models that can reason about plausible diverging future paths and non-linear motion.

## 2.2 Trajectory Forecasting

Pedestrian trajectory forecasting has been studied independently of the closely related task of object tracking. Forecasting is challenging because (i) human behavior and, therefore, future motion underlies the effect of complex social and scene interactions and latent navigating intent. Moreover, (ii) entire scene geometry is usually not directly visible to the observer, and in general, it is difficult to localize past trajectories precisely. To this end, existing models use standard datasets [53, 35, 38, 55] and study forecasting in idealized conditions: given an accurate bird's-eye view of the scene and perfectly-localized past trajectories to predict trajectory continuations in metric space.

**Quantifying forecasting accuracy.** Forecasting performance is measured in metric space as $L_2$ distance between the prediction and ground-truth trajectory (as final displacement error, FDE, or average displacement error, ADE) *wrt.* top-k forecasts (commonly $k = 20$). We note that this approach mainly incentivizes high forecasting recall and neglects forecasting precision which is important for the application of forecasting methods [13].

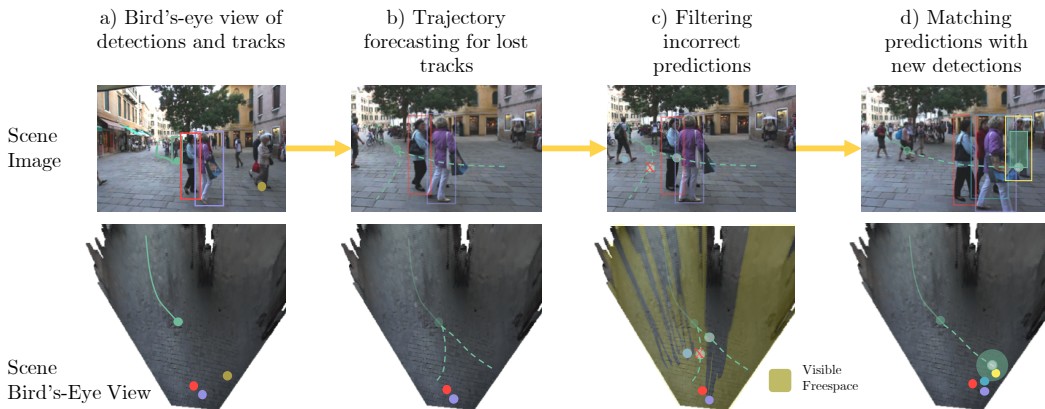

| a) Bird's-eye view of detections and tracks | b) Trajectory forecasting for lost tracks | c) Filtering incorrect predictions | d) Matching predictions with new detections |

Figure 2: **Our method:** we bridge long-term occlusions by (a) localizing object tracks in bird's-eye view via the estimated homography and (b) forecasting future trajectories for *lost* tracks. We (d) continually aim to match these *inactive* track predictions with new object detections and remove incorrect predictions under a visibility constraint (c).

**Prior work.** Early forecasting methods were deterministic, firstly based on physical models [25], and later on data-driven LSTM-based encoder-decoder networks [1] methods, focusing on modeling social [23, 1, 2] and scene [56, 31] interactions. The forecasting task is inherently uncertain, and we need to express the stochasticity in the model. With learning a distribution of possible future trajectories, generative models [23, 56, 31, 2, 14, 13] have emerged as state-of-the-art prediction methods. Recent efforts have been explicitly focusing on conditioning forecasting on estimated pedestrian goal/intent [14, 43, 42] and estimating multimodal posterior distributions [38, 13] that yield diverse trajectories that cover different plausible directions. These deep neural network approaches can model complex and non-linear trajectories beyond simple linear models.

Can we bring the *two worlds* together, and if so, *how*? Furthermore, which of the aforementioned aspects of forecasting methods (*i.e.*, stochasticity, non-linearity, multimodality, diversity, accounting for interactions) are *important* in the context of multi-object tracking? These are the questions we discuss in the following sections.

## 3   Methodology

In this section, we present our method for long-term multi-object tracking based on trajectory forecasting in bird's-eye view (BEV) scene representation. Simply applying trajectory prediction to multi-object tracking is not trivially possible, as object trajectories observed in the image space break multiple assumptions of real-world trajectory prediction. While trajectory prediction works in bird's-eye view coordinates, the motion and size of objects in image space depend on the camera's intrinsic parameters, orientation, and position. In addition, we face temporal (limited length of observation), association (association errors along with observation), and measurement (imprecise localization of objects) uncertainties of the trajectories. Contrarily, objects are represented as bounding boxes in the image instead of single 2D positions for the object tracking task. To bridge the gap between prediction and tracking, we must find a transformation from the image to the real space. We assume objects move on a planar ground to formulate such a transformation. Thus, the bottom-center points of detection bounding boxes $p$ can be mapped to a 2D BEV coordinate $x$ via an initially unknown homography transformation $H$ that relates the homogeneous coordinates as $x \propto H \cdot p$.

**Overview.** Given a monocular video sequence captured from a stationary camera from *arbitrary* viewpoint, we first estimate the homography $H$, which maps the image plane to the 3D world ground-plane for the whole sequence (Section 3.1). Then, we incorporate our model into an online tracker that takes a monocular tracker output and localizes tracks and detections in BEV space (Figure 2*a*) using the estimated homography. Next, we forecast lost tracks in BEV space (Figure 2*b*) using our trajectory forecasting network (described in Section 3.2). Finally, we integrate forecasts into the online tracker (Section 3.3) while accounting for the uncertainty in estimated forecasts, and match new detections to existing tracks to resolve short- and long-term occlusions (Figure 2*d*).

### 3.1   Data-driven Homography Estimation

For combining monocular object tracking and forecasting, we first need to transform object detections and tracks from the image sequence into points and trajectories in a bird's-eye view representation.

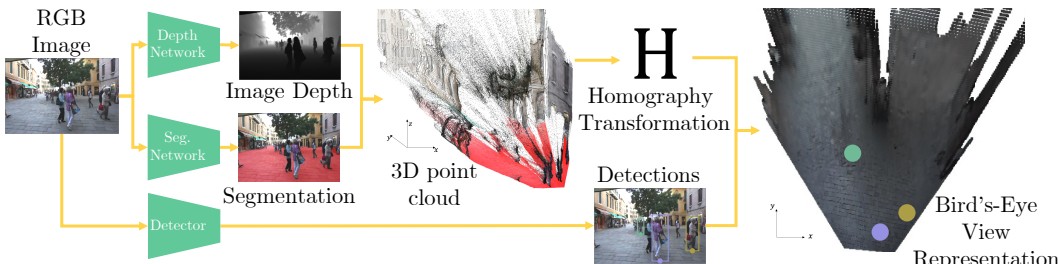

Figure 3: We estimate the homography $H$ for a sequence by reconstructing a 3D point cloud using a monocular depth estimator. We obtain ground image-to-point-cloud correspondences using a semantic segmentation model that masks ground pixels as needed to estimate the homography matrix. With the estimated homography matrix, we transform the bottom points of bounding boxes to 2D BEV coordinates.

Given a set of 2D object detections represented as bounding boxes localized in the image plane, we aim to find a homography $H$ that maps their bottom-center positions to their corresponding 2D BEV coordinates. In Figure 3, we outline our homography estimation method. We first train a monocular depth estimator [6] on a synthetic dataset [19] to reconstruct a 3D point cloud (with estimated or known intrinsics) of the first frame of a static sequence. Then, we leverage the semantic segmentation network [73] to mask, select, and fit a plane to the ground pixels. We estimate the normal vector of the ground plane in 3D and align the plane to the $XY$ plane. Then, we project ground points along the $z$-axis, leaving us with a pairwise correspondence between ground pixels in the image and a 2D position in BEV, as needed to estimate the homography between the two planes. We also linearize the homography transformation for pixel positions close to the plane's horizon to prevent the transformation from diverging (for more information, see Appendix A.1).

**Static camera.** We compute the homography only for the first frame of the sequence and use it throughout the sequence, making our pedestrian localization robust to temporal fluctuations of the depth estimator.

**Moving camera.** For moving camera sequences, we also need to account for the egomotion of the camera, which we estimate between consecutive frames as follows. First, we compute a frame-dependent homography $H_t$ for each frame. Then, we compute pairwise pixel-correspondences between (masked) ground pixels using optical flow [12] and compute a translation vector between the two point sets (lifted to 3D via $H_t$).

Empirically, we observe that estimating only translation (without rotation) yields more robust egomotion estimates.

### 3.2 Forecasting

Localization of object tracks in BEV enables us to leverage data-driven forecasting models beyond simple linear motion to reason for future trajectories. However, these models expect ground-truth fixed-size past trajectory observations, while our projected trajectories are noisy and of varying lengths. As discussed in Section 2.2, prediction models are optimized for metrics that incentivize a large number of predictions and minimize $L_2$ distance to ground-truth trajectories. It is thus unclear how different proposed concepts translate into real-world tracking scenarios. We, therefore, identify the main design patterns proposed in the forecasting community and verify their impact *directly* in the context of *forecasting to track*.

**Preprocessing.** Forecasting models encode trajectories using an LSTM encoder-decoder [1] architecture, which takes a fixed-size observed trajectory as input and predicts a future trajectory. We construct input trajectories from temporally consecutive detections of the same track ID localized in BEV. To account for localization noise, we smooth the noisy observations using the Kalman filter and linearly extrapolate trajectories into the past to get trajectories of the required fixed-size input length.

**Trajectory forecasting design patterns.** In our experimental setup, we build on the LSTM encoder-decoder architecture [1] and include the following key design patterns recently emerging in the forecasting community.

*Stochasticity.* Stochastic trajectory predictors enable us to sample multiple plausible future trajectories to account for the uncertainty in future positions. We follow the approach by [23] and learn a generative GAN [22] model and train it with a best-of-many [7] loss. As a result, the network

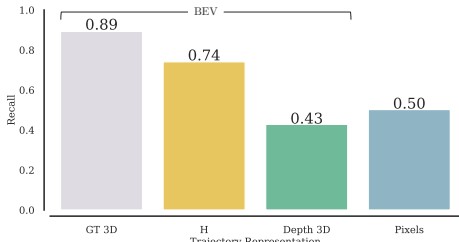 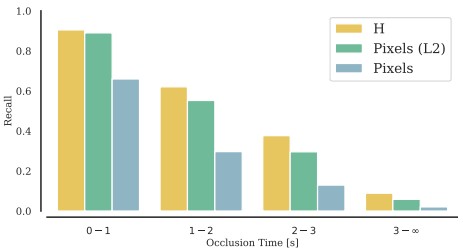

(a) Overall recall (BEV and pixel-space).   (b) Recall *wrt.* different occlusion lengths.

Figure 4: Endpoint matching recall of predictions and GT trajectories using a linear motion predictor. A prediction is successfully matched when $\Delta_{\text{IoU}} > 0.5$ or $\Delta_{L_2}$ distance $< 2m$. We also project the prediction back to the image for forecasts in the bird's-eye view. The model *Pixel (L2)* predicts motion in pixel space and projects the endpoint into BEV for matching.

internally learns an observation-conditioned distribution of future trajectories, from which we can sample.

*Social Interactions.* Social interactions impact future motion: pedestrians adapt their trajectories on-the-fly to avoid collisions. Several methods [23, 1, 2, 31] account for interactions in the forecasting literature. These methods leverage pooling [23], attention [56], or graph neural netwchecklorks [31] to provide social context (*i.e.*, trajectories of surrounding agents) to the trajectory decoder. To answer whether modeling social interactions matters for tracking, we implement Social GAN (S-GAN) [23], which uses pairwise interaction features between neighboring pedestrians by using max-pooling before they are passed to the decoder.

*Multimodality and Diversity.* While the aforementioned generative models learn a distribution over trajectories, they need to sample many trajectories to cover all modes present in the scene, as learning a multimodal posterior with a single GAN is difficult [64]. To predict the scene's main modes with as few samples as possible, we implement a multi-generator GAN network [13], extending the presented GAN architecture by training multiple decoder heads, where each decoder learns to focus on a particular model. As a result, we get a set of plausible but maximally separated predictions by sampling from these different generators.

### 3.3 Tracking via Forecasting

We assume we have an online object tracker capable of tracking visible objects (*e.g.*, bounding box regression-based tracker [4]). As long as tracks are being updated with new detections, we consider them *active* and keep them in the active set $\mathcal{S}_A$. Once we cannot associate a detection, a track becomes *inactive* and is stored in the set of inactive tracks $\mathcal{S}_I$. For each frame $t$ the tracker outputs a set of tracks $\mathcal{O} = (o_1, \ldots, o_M)$ with $o_i = (\text{ID}_i, b_i, f_i)$ where $\text{ID} \in \mathbb{N}^+$ represents the track identity, $b \in \mathbb{R}^4$ denotes a bounding box in pixel space (see Figure 2a), and $f \in \mathbb{R}^D$ represents a $D$-dim feature vector encoding the appearance information obtained from a pre-trained convolutional network [24]. We localize bounding boxes in BEV coordinates $x \in \mathbb{R}^2$ using our estimated homography $H$.

**Quo Vadis?** If an object track becomes inactive (*i.e.*, temporally lost), we move the active track into the memory bank and predict $k$ trajectories of length $\tau_{max}$ in BEV space using the trajectory forecasting model as described in Section 3.2. As long as the track is inactive and not yet matched, we move along the predicted trajectory and do not predict an entirely new trajectory in each frame.

**Filtering and removing predictions.** *No prediction can live forever.* When we use stochastic trajectory predictors with multiple predictions, we need to limit the number of inaccurate or obsolete predictions to decrease the chance of false re-association. In practice, we limit the lifetime of a prediction to a maximal lifetime of $\tau_{max}$ and try to filter out *unlikely* forecasts. We use spatial and social context to determine the *freespace* [29] in which objects should be visible to the camera. We assume that visible objects eventually are detected and, therefore, remove prediction branches that should be visible for more than $\tau_{vis}$ frames.

We consider an object as *visible* if neither scene nor other pedestrians occlude the object. In particular, this means that the predicted BEV position lies in an area of the projected ground mask (shown in Figure 2c) and has no bounding box overlap $\geq 0.25$ with any other object detection, closer to the camera. Relative order can be determined based on bottom bounding box coordinates for amodal detections. If all predictions from an inactive track are removed even before $\tau_{max}$, we also remove the entire track.

**Matching predictions with new detections.** Given the trajectory forecasts, we match them with new detections via bi-partite matching, following the standard practice in tracking [81, 68, 39, 5]. This boils down to computing association costs $c_{ij}$ between the predictions of an inactive track $i$ and new un-associated detections $j$:

$$c_{ij} = (\Delta_{\text{IoU}} + \max(\tau_{L_2} - \Delta_{L_2}, 0)) \cdot (\Delta_{\text{App}} \geq \tau_{\text{App}} \text{ and } \Delta_{\text{IoU}} \geq \tau_{\text{IoU}}), \quad (1)$$

where $\Delta_{\text{IoU}}$ is the IoU score between the two bounding boxes, $\Delta_{L_2}$ is the Euclidean distance between the prediction $i$ and a detection $j$ in BEV, and $\Delta_{\text{App}}$ represents the cosine distance between visual features $f_i$ and $f_j$. $\tau_{L_2}, \tau_{\text{IoU}},$ and $\tau_{\text{App}}$ denote thresholds for the matching. Therefore, we determine an association in BEV metric space and in the image domain using IoU bounding box overlap between the forecast and detected box.

Matching tracks based on spatial distance in real space leads to high recall and reduces the number of ID switches (IDSW) but may also lead to an increase in ID transfer errors (IDTR), especially in crowded scenes with many new detections close to each other. While forecasting significantly narrows combinatorial search space for associations, verifying potential associations with an appearance model is still beneficial in practice. To decrease the number of wrong associations, we require a minimal visual similarity $\tau_{\text{App}}$ and minimum IoU overlap of the bounding boxes $\tau_{\text{IoU}}$ for close objects. These thresholds serve as a filter of visually incompatible matches but do not add to the value of the cost function for the matching. In essence, we obtain a pre-selection of potential matching candidates by using the trajectory forecast and filter those if the appearance drastically deviates between the last observation and the new detection.

## 4 Experimental Evaluation

In this section, we first discuss our evaluation test-bed, followed by an experimental study on bird's-eye-view (BEV) trajectory reconstruction (Section 4.1). Then, we analyze different forecasting design patterns applied to the domain of object tracking in BEV space and discuss the relevance of different model modules for tracking (Section 4.2). Afterward, we demonstrate how our approach can be used to improve several vision-based MOT methods on static sequences and to justify our design decisions. Finally, we show that our forecasting model can be used to establish new state-of-the-art on the real-world *MOT17* and *MOT20* datasets (Section 4.4). For visualization of our tracking method, we refer the reader to Appendix C.

**Datasets.** We evaluate our trajectory prediction framework on different publicly available pedestrian tracking datasets, namely synthetic *MOTSynth* [19] and two real-world *MOT17* and *MOT20* datasets [15]. *MOTSynth* is a large synthetic dataset for multi-object tracking. It provides 764 diverse sequences with various viewpoints, lighting, and weather conditions. Importantly, it provides ground-truth depth information and 3D key points for pedestrians, allowing us to study the suitability of different methods for BEV trajectory reconstruction. *MOT17* [47] and *MOT20* [16] are real-world tracking datasets commonly used to benchmark pedestrian tracking models. We use these datasets to evaluate our method on real-world recordings. For our experiments, we utilize the commonly used split of the *MOT17* training set, where the first half of each sequence is used for training and the second half for the evaluation [37, 80, 72].

**Metrics.** For measuring the quality of the bird's-eye view reconstruction, we indirectly evaluate the quality of the reconstruction by evaluating the forecasting and tracking performance.

To compare different models for trajectory forecasting, we report the standard $L_2$ final displacement error (FDE) for top-k predictions for $2s$ and $4s$ prediction horizons (see Section 2.2).

For multi-object tracking evaluation, we report higher-order tracking accuracy (HOTA) [41], with a focus on the association aspect of the task. To this end, we also report AssA, AssPr, and the number of ID switches IDSWs. Additionally, we report IDSW when the tracker loses an object and re-initiates a new track for the same object when it re-appears. We call these errors as ID$^{\text{lost}}$. For metric discussion, we refer to Section 2.1. Metrics labeled with either $S$ (short) or $L$ (long) only consider prediction or occlusion lengths shorter or longer than $2s$, respectively

**Hyperparameters.** For all experiments, we use the following parameters for the matching of detections with inactive tracks: $\tau_{L_2} = 2.5m, \tau_{\text{App}} = 0.8, \tau_{\text{IoU}} = 0.2$ The maximal lifetime of prediction is $\tau_{max} = 6s$ and maximal visibility $\tau_{\text{vis}} = 1s$ before it is removed. We refer the reader to Appendix B for further information on implementation details.

**Object trackers.** We study and ablate our method on eight high-ranked state-of-the-art trackers of *MOTChallenge* and refer to them as *baseline*. We use BYTE [80], JDE [68], CSTrack [37], FairMOT [81], TraDes [72], QDTrack [51], CenterTrack [82] and TransTrack [63] for an evaluation

Table 1: Which forecasting modules matter for tracking? Evaluated on *MOT17* validation set.

| Model | Nr. Samples | Deterministic | Stochastic | Social | Multi-modal | Prediction FDE$_S$ ↓ | FDE$_L$ ↓ | Tracking HOTA ↑ | AssA ↑ | AssRe ↑ | AssPr ↑ | ID$_S^{lost}$ ↓ | ID$_L^{lost}$ ↓ |
|---|---|---|---|---|---|---|---|---|---|---|---|---|---|
| Baseline | | | | | | – | – | 50.71 | 46.87 | 51.80 | **78.11** | 0 % | 0 % |
| Static | 1 | ✓ | | | | 1.59 | 2.09 | 53.84 | 53.51 | 60.04 | 72.95 | -14.77 % | -8.40 % |
| Kalman Filter (pixel) | 1 | ✓ | | | | – | – | 54.08 | 54.02 | 60.45 | 72.81 | **-22.37 %** | -8.99 % |
| Kalman Filter | 1 | ✓ | | | | 0.69 | 1.23 | 54.11 | 54.04 | 60.75 | 71.73 | -19.50 % | -16.07 % |
| GAN | 3 | | ✓ | | | 0.85 | 1.26 | 54.43 | 54.61 | 61.11 | 73.21 | -17.99 % | -8.64 % |
| GAN | 20 | | ✓ | | | **0.65** | **0.99** | 53.81 | 53.40 | 60.45 | 71.31 | -18.03 % | -15.63 % |
| S-GAN | 3 | | ✓ | ✓ | | 0.87 | 1.21 | **54.52** | 54.78 | 61.22 | 73.28 | -16.92 % | -8.57 % |
| MG-GAN | 3 | | ✓ | | ✓ | 0.67 | 1.03 | **54.52** | **54.80** | **61.35** | 73.13 | -21.19 % | **-17.43 %** |

on the *MOT17* validation set and BYTE and CenterTrack on the *MOT20* training dataset. These trackers use ReID similarity and/or simple motion cues for bridging short-term occlusions.

## 4.1 Bird's-Eye View Estimation

This section discusses different approaches to obtaining scene BEV representations of detected objects in the image for forecasting. We work with static sequences of the *MOTSynth* dataset (that provides depth maps used for evaluating and training a monocular depth estimator). With this, we test a linear motion model to gap occlusions of different durations, which we obtain by running a CenterTrack [82] baseline tracker. We evaluate the ratio of successful matches between the target and the linear prediction and count a match to be successful if the IoU of the predicted bounding box is larger than $0.5$ or the $L_2$ distance in metric space is lower than $2m$. We get the predicted bounding box by translating the last observed bounding box by the predicted displacement in the image.

**Baselines.** We compare motion in (i) BEV and (ii) pixel space. We evaluate different approaches to localize trajectories: (a) using ground truth (GT) 3D coordinates orthographically projected to BEV (oracle), (b) the proposed homography estimation as described in Section 3.1, and (c) directly using learned monocular depth estimates and resulting point clouds, followed by orthogonal projection of points these representing an object.

**Conclusions.** As seen in Figure 4, GT 3D (oracle) based motion estimates solve $89.3\%$ of the gaps, suggesting that the motion in the synthetic dataset is dominantly linear. Our proposed data-driven homography estimation approach only drops by $15\%$ compared to using ground-truth 3D keypoints. By contrast, estimating linear motion in pixels space only resolves $50.2\%$, and using per-frame monocular depth estimates $43.1\%$ of the occlusion gaps. This is likely because such depth estimates are not temporally stable. As can be seen in Figure 4b, this performance is especially apparent for longer occlusion gaps. Furthermore, we forecast motion in pixel space but transform the prediction into BEV for matching. While increasing performance compared to exclusive forecasting and matching in pixel space, we find that the results are inferior to predictions in BEV due to the distortion of projecting real motion into the image plane. We conclude that our proposed homography transformation is suitable for forecasting.

## 4.2 Trajectory Prediction Models

In this section, we evaluate different forecasting models and components (as discussed in Section 3.2). We compare a constant-velocity model (Kalman filter) in BEV and pixel space, an identity (static) model, and a stochastic GAN predictor generating $k = 3$ and $k = 20$ samples. Furthermore, we test predicting social interactions with GAN and the multimodal trajectories with MG-GAN in BEV space.

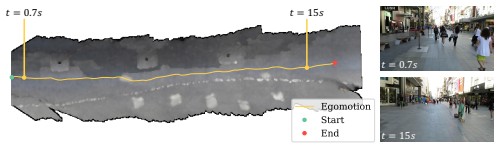

Figure 5: Visualization of BEV reconstruction for moving camera sequence and egomotion estimation.

Table 2: Ablation of matching prediction and effect of different thresholds $\tau$ on different tracking metrics.

| Scores $L_2$ | IoU | Threshold $\tau_{IoU}$ | $\tau_{App}$ | HOTA ↑ | AssA ↑ | AssRe ↑ | AssPr ↑ | ID$^{lost}$ ↓ |
|---|---|---|---|---|---|---|---|---|
| ✓ | | | | 53.89 | 53.56 | 60.43 | 72.21 | -16.18 % |
| ✓ | | | ✓ | 53.89 | 53.57 | 60.51 | 71.69 | -16.26 % |
| ✓ | | ✓ | ✓ | 54.10 | 53.92 | 60.43 | **73.36** | -16.84 % |
| | ✓ | | | 54.13 | 54.01 | 60.97 | 72.00 | -24.06 % |
| ✓ | ✓ | | | 53.75 | 53.35 | **61.17** | 69.27 | **-28.02%** |
| ✓ | ✓ | ✓ | | 53.97 | 53.75 | 61.08 | 70.73 | -26.93 % |
| ✓ | ✓ | | ✓ | 54.06 | 53.92 | 61.07 | 71.01 | -21.40 % |
| ✓ | ✓ | ✓ | ✓ | **54.27** | **54.29** | 61.08 | 72.36 | $-20.53\%$ |

**Forecasting.** We observe in Table 1 that the linear model performs well for short-term windows ($0.69$ FDE$_S$), suggesting that linear motion is suitable for short occlusions. We also do not find a significant difference between GAN w/o social module (S-GAN). While FDE error suggests vanilla GAN ($k = 20$) yields the best forecasting results ($0.65$ FDE$_S$ and $0.99$ FDE$_L$), but this configuration

Table 3: We improve tracking results of *all* top-8 state-of-the-art models (MOT17 validation set and *MOT20* training set). Differences to the baseline performance are shown in $(\cdot)$.

| | MOT17 (val, static scenes) | | | | | | | | MOT20 (train) | |
|---|---|---|---|---|---|---|---|---|---|---|
| | BYTE [80] | CenterTrack [82] | CSTrack [37] | FairMOT [81] | JDE [68] | TraDeS [72] | TransTrack [63] | QDTrack [51] | BYTE [80] | CenterTrack [82] |
| HOTA | 71.36 (+0.21) | 61.78 (+3.56) | 61.60 (+0.43) | 58.42 (+0.09) | 51.06 (+0.20) | 62.45 (+0.67) | 60.68 (-0.23) | 58.87 (+0.54) | 56.85 (+0.06) | 32.71 (+0.62) |
| AssA | 73.96 (+0.49) | 66.18 (+7.54) | 63.84 (+0.8) | 59.21 (+0.37) | 54.36 (+0.45) | 67.41 (+1.6) | 63.47 (-0.49) | 60.14 (+1.22) | 53.97 (+0.20) | 28.94 (+1.34) |
| AssRe | 79.21 (+0.66) | 69.66 (+8.38) | 69.15 (+1.07) | 64.31 (+0.5) | 60.82 (+0.89) | 73.1 (+2.43) | 69.19 (+0.02) | 65.31 (+2.06) | 59.89 (+0.4) | 34.34 (+5.12) |
| AssPr | 83.11 (-0.67) | 81.75 (-5.47) | 77.79 (-2.28) | 74.45 (-1.71) | 80.9 (-2.35) | 80.0 (-1.91) | 79.53 (-1.68) | 77.4 (-2.98) | 68.65 (-5.24) | 52.37 (-21.06) |
| IDSW | 84 (-3) | 137 (-146) | 269 (-28) | 198 (-12) | 316 (-19) | 106 (-32) | 112 (-3) | 219 (-34) | 1815 (-78) | 5240 (-2700) |
| MOTA | 80.09 (+0.01) | 70.77 (+0.39) | 71.31 (+0.05) | 71.82 (+0.05) | 59.57 (+0.06) | 70.93 (+0.09) | 69.5 (+0.01) | 69.61 (+0.08) | 73.38 (+0.0) | 47.57 (+0.24) |
| IDF1 | 82.92 (+0.42) | 74.46 (+7.13) | 74.16 (+0.95) | 73.93 (+0.59) | 65.01 (+1.27) | 76.36 (+1.21) | 71.46 (+0.02) | 70.41 (+0.77) | 72.47 (+0.37) | 45.85 (+4.13) |
| IDR | 78.61 (+0.39) | 65.25 (+6.25) | 67.53 (+0.87) | 66.23 (+0.53) | 56.08 (+1.09) | 67.12 (+1.06) | 61.39 (+0.01) | 62.17 (+0.68) | 66.44 (+0.34) | 35.87 (+3.23) |
| IDP | 87.72 (+0.44) | 86.71 (+8.3) | 82.23 (+1.05) | 83.65 (+0.67) | 77.31 (+1.51) | 88.55 (+1.4) | 85.47 (+0.02) | 81.17 (+0.89) | 79.7 (+0.41) | 63.53 (+5.72) |

leads to the lowest association precision (71.31) and HOTA score (53.81) *wrt.* tracking performance. This result suggests a misalignment of evaluation metrics used in forecasting and tracking; a better forecaster in terms of ADE/FDE does not necessarily lead to a better tracker. This is a known drawback of ADE/FDE metrics, which essentially measure only recall and not the precision of the forecasting output. Furthermore, this shows the careful trade-off between the number of predictions $k$ and the recall/precision of the predictions and tracking results.

**Tracking.** To investigate the effect on long-term occlusions, we focus on the change of $\text{ID}_L^{lost}$ for short ($t_{occl} \leq 2s$) and long ($t_{occl} > 2s$) occlusion gaps. As can be seen in Table 1, even the static motion model solves $8.4\%$ ($\text{ID}_L^{lost}$), as many occluded objects do not move. By modeling linear motion (Kalman filter in pixel space), we can improve short-term re-association for $0.59pp$ (long-term IDSW) over the static model. We focus the discussion on long occlusion gaps. In terms of the generative model, we observe that interaction-aware S-GAN ($8.64\%$ $\text{ID}_L^{lost}$) is on-par with vanilla GAN ($8.57\%$ $\text{ID}_L^{lost}$) for $k = 3$; interestingly, both are below linear Kalman filter (BEV) performance, suggesting that these models suffer from low precision. Only MG-GAN, explicitly trained to generate multimodal trajectories, outperforms the linear model ($17.43\%$ $\text{ID}_L^{lost}$) and significantly outperforms vanilla GAN with only three samples. These conclusions generalize to tracking metrics.

### 4.3 Tracking Evaluation

In this section, we study the impact of forecasting models on the valuation set's tracking performance. First, we discuss the impact of different design decisions on matching strategy, as explained in Section 3.3.

**Trajectory matching.** First, we ablate the matching cost function (Equation (1)). As can be seen in Table 2, we find that a combination of $L_2$ and IoU without any threshold $\tau$ leads to the highest decrease in terms of $\text{ID}^{lost}$ ($-28.02\%$) and overall highest association recall (AssRe) (61.17). However, this is at the cost of decreasing association precision (AssPr) ($-4.09$). We obtain the highest AssPr (73.36) by only relying on $L2$ matching and thresholding, however, at the loss of AssRe ($-0.74$). Adding appearance-based $\tau_{\text{App}}$ and IoU $\tau_{\text{IoU}}$ thresholds provide the best trade-off and overall highest AssA (54.29) and HOTA score (54.27) while still recovering $21\%$ of lost trajectories.

**Validation results.** In Table 3, we present the performance of different state-of-the-art trackers on the *static sequences* of *MOT17*-val and *MOT20*-train (trained on *MOT17*), equipped with our trajectory forecasting model. As can be seen, our model brings stable improvements over all the key metrics: HOTA, AssA, and IDSW. Our trajectory prediction model consistently reduces IDSW for all models. This is also shown in Figure 1b where we demonstrate that our forecasting model improves ID recall significantly for occlusion times $> 1s$.

While our focus was on sequences with stationary viewpoints, we show that our model is also applicable to sequences with moving cameras by estimating the camera's egomotion as described in Section 3.1. In Table 4, we present results on the *moving sequences* of the MOT17 validation set (excluding sequence MOT17-05 for which the quality and consistency of our depth estimator was too low to construct time-consistent homographies). As can be seen, we improve 6 out of 8 trackers *wrt.* HOTA score and even improve CenterTrack [82] by 3.07 pp. We visualize the traversed BEV map of sequence MOT17-07 in Figure 5.

### 4.4 Benchmark Evaluation

In this section, we apply our method to state-of-the-art tracker ByteTrack [80] and, by improving its long-term tracking capabilities, establish a new state-of-the-art on the *MOT17* & *MOT20* benchmarks. We evaluate our method in the *private detection* regime, as these trackers use private detectors.

In Table 5, we compare our *QuoVadis* to the base tracker ByteTrack [80] and compare both to *MOT17* benchmark published top-performers. We improve performance on key metrics overall top methods. Notably, we reduce the number of identity switches by 93 compared to [80] and establish a new

Table 4: Results of top-8 state-of-the-art models on dynamic scenes of the MOT17 validation set excluding MOT17-05. Differences in the baseline performance are shown in (·)

| | MOT17 (val, moving scenes) | | | | | | | |
| --- | --- | --- | --- | --- | --- | --- | --- | --- |
| | BYTE [80] | CenterTrack [82] | CSTrack [37] | FairMOT [81] | JDE [68] | TraDeS [72] | TransTrack [63] | QDTrack [51] |
| HOTA | 60.08 (+0.02) | 51.77 (+3.07) | 54.51 (0.0) | 56.1 (0.0) | 52.14 (+1.47) | 53.36 (+1.27) | 52.7 (+0.28) | 52.28 (+0.76) |
| AssA | 60.44 (+0.03) | 53.18 (+6.49) | 59.04 (+0.0) | 61.15 (0.0) | 55.32 (+3.06) | 54.08 (+2.44) | 51.99 (+0.54) | 53.71 (+1.6) |
| AssRe | 66.53 (-0.0) | 58.21 (+8.32) | 63.35 (+0.0) | 65.87 (0.0) | 61.52 (+3.75) | 59.86 (+3.1) | 59.13 (+0.79) | 61.8 (+3.82) |
| AssPr | 78.29 (+0.09) | 76.98 (-4.66) | 80.46 (-0.0) | 79.21 (0.0) | 73.5 (-0.98) | 75.52 (-3.08) | 72.45 (-0.32) | 72.53 (-4.76) |
| IDSW | 54 (+1) | 131 (-62) | 97 (-5) | 86 (0) | 122 (-10) | 99 (-11) | 120 (-5) | 71 (-7) |
| MOTA | 72.54 (-0.01) | 59.46 (+0.46) | 60.68 (+0.04) | 63.78 (0.0) | 60.52 (+0.07) | 64.13 (+0.08) | 63.64 (+0.04) | 60.21 (+0.05) |
| IDF1 | 73.11 (0.0) | 63.48 (+5.76) | 70.69 (0.0) | 73.1 (0.0) | 68.23 (+1.85) | 67.72 (+2.29) | 64.08 (+0.79) | 65.81 (+2.86) |

Table 5: Comparison under the "private detector" protocol on *MOT17* test set.

| Tracker | HOTA | IDF1 | MOTA | IDSW | AssA |
| --- | --- | --- | --- | --- | --- |
| ReMOT [76] | 59.73 | 71.99 | 77.01 | 2853 | 57.08 |
| CrowdTrack [61] | 60.26 | 73.62 | 75.61 | 2544 | 59.26 |
| TLR [67] | 60.72 | 73.58 | 76.48 | 3369 | 58.88 |
| MAA [60] | 61.98 | 75.88 | 79.36 | 1452 | 60.16 |
| ByteTrack [80] | 63.05 | 77.30 | 80.25 | 2196 | 61.97 |
| QuoVadis (Ours) | **63.14** | **77.71** | **80.27** | **2103** | **62.07** |

Table 6: Comparison under the "private detector" protocol on *MOT20* test set.

| Tracker | HOTA | IDF1 | MOTA | IDSW | AssA |
| --- | --- | --- | --- | --- | --- |
| FairMOT [81] | 54.42 | 68.44 | 59.57 | 1881 | 56.6 |
| CrowdTrack [61] | 54.95 | 68.24 | 70.68 | 3198 | 52.57 |
| MAA [60] | 57.28 | 71.15 | 73.90 | 1331 | 55.14 |
| ReMOT [76] | 61.15 | 73.14 | 77.42 | 1789 | 58.68 |
| ByteTrack [80] | 61.34 | 75.20 | 77.76 | 1223 | 59.55 |
| QuoVadis (Ours) | **61.48** | **75.70** | **77.77** | **1187** | **59.87** |

state-of-the-art in terms of HOTA (63.14). We observe similar trends on *MOT20*, where we improve over the base tracker ByteTrack [80] by +0.5 in terms of IDF1 and reduce the number of identity switches by 36, similarly establishing a new state-of-the-art (61.48 HOTA).

## 5 Remarks and Limitations

The paper primarily focuses on the conceptual work of building an entire pipeline from video to tracks studying different forecasting paradigms, and showing the benefit of leveraging trajectory forecasting in BEV for the tracking task. Nevertheless, we want to outline further remarks and limitations of our work.

**Model complexity.** Our model is complex, consisting of multiple sub-modules, as constructing object trajectories in BEV space based on a monocular video and trajectory forecasting are challenging problems. We foresee that future work will improve the end-to-end integration and efficiency of the algorithms.

**Bird's-eye view reconstruction.** A vital part of our approach is an accurate homography transformation that allows us to project the objects in the image into the 3D ground plane. However, the homography transformation depends on the quality of depth estimates, which makes the overall approach sensitive to errors in 3D localization. Future work will benefit from further development of time-consistent depth estimators. Furthermore, the presented trajectory prediction models do not yet account for the BEV localization uncertainty, which results from errors in the transformation or the simplified assumption of the ground plane. These limitations show the need to develop trajectory forecasting models that account for the localization uncertainties of the upstream tasks.

## 6 Conclusion

This paper presents a study on how to bridge the gap between real-world trajectory prediction and single-camera tracking. Throughout our paper, we identified challenges and solutions to leveraging real-world trajectory prediction to benefit single-camera tracking. In particular, we focus on resolving the re-identification of objects after long-term occlusions. Here, we start from the first principles, questioning motion representation in pixel space and using a combination of models to construct a more accurate BEV representation of the scene. We find that the key component is a forecasting approach reasoning about multiple feasible future directions with a small set of multimodal forecasts. We can substantiate our conclusion by achieving new state-of-the-art performance on the *MOT17* and *MOT20* datasets.

Ultimately, we have showcased a novel way of combining state-of-the-art trajectory prediction models and multi-object tracking task. We have outlined a new way of thinking about motion prediction in tracking and motivating the beneficial symbiosis of both tasks. We hope that both fields start moving towards each other and incorporate the requirements and needs of each other.

**Acknowledgments.** This research was partially funded by the Humboldt Foundation through the Sofja Kovalevskaja Award.

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
