# OpenReview forum: "Quo Vadis: Is Trajectory Forecasting the Key Towards Long-Term Multi-Object Tracking?"
_NeurIPS.cc/2022/Conference — NeurIPS 2022 Accept_

### Official Review · Reviewer_Tgjz · 2022-07-09

**Rating:** 6
**Confidence:** 4
**Soundness:** 3 good
**Presentation:** 3 good
**Contribution:** 3 good

**Summary:**

This paper discovers that trajectory predictions for moving agents will significantly reduce this search space and explores the trajectory prediction can improve long-term tracking robustness of MOT. Furthermore, they show that their proposed method reasons MOT in the bird-eye space and generates a small yet diverse set of forecasts while accounting for their localization uncertainty. Therefore, they manage to advance state-of-the-art trackers on the public benchmarks.



**Questions:**

- The proposed model estimates the depth and segmentation masks of the video frame, and thus project to BEV space. The quality of BEV is the basis of trajectory prediction. This paper indeed investigated how good the view projection (in Fig. 4) in synthetic data. I wonder if is it possible to conduct experiments to assess the impact of different view projection methods on MOT?
- The pointers to tables are missing, which makes reading a bit difficult.
- MOTA seems to be a more popular metric in MOT challenge. In previous MOT works (e.g., Bytetrack), they provide MOTA metrics. For better reference, it would be better to provide MOTA as well.
- The implementation details on how to cope with the moving camera are not clearly provided.

**Ethics Review Area:**

["Privacy and Security (e.g., consent)"]

**Limitations:**

- The major limitation of this work on the uncertainty of homography transformation has been discussed. Their proposed method partially account for these uncertainties via simple strategies. Considering this is a novel exploration along this direction, I think the limitations have been sufficiently addressed.

- The potential negative societal impact of their work was not dicussed.

**Strengths And Weaknesses:**

### Strengths
- This idea is interesting and it will benefit the development of MOT community. Considering occlusions are one of the main challenges in tracking, this work proposes a novel way to overcome this concern.
- This paper identifies several hurdles in the integration of trajectory prediction and MOT. Besides, there are several interesting and inspiring conclusions.
- The presentation of this paper is fine and the organization is clear.
- They achieve the state-of-the-art performance on public benchmarks. Maybe, it may inspire more following MOT methods on this path.

### Weaknesses
- The framework seems too complex. It includes five independent sub-models, e.g., depth estimation network and segmentation network. All of these sub-models require separate training, which may degrades the robustness of the framework.
- Before the deep learning era, there are a few works that already attempted to incorporate crowd motion prediction or crowd motion models, e.g., social force, to MOT, some of which are listed below. It'd be better to refer to these works.
> G Antonini，SV Martinez，M Bierlaire，JP Thiran: _Behavioral Priors for Detection and Tracking of Pedestrians in Video
Sequences_ IJCV 2006
> Stefano Pellegrini, Andreas Ess, Luc Van Gool: _You'll Never Walk Alone: Modeling Social Behavior for Multi-target Tracking_ ICCV 2009
> Kota Yamaguchi, Alexander C. Berg, Luis E. Ortiz, Tamara L. Berg: _Who are you with and Where are you going?_ CVPR 2011
> Wenxi Liu, Antoni B. Chan, Rynson W. H. Lau, Dinesh Manocha: _Leveraging Long-Term Predictions and Online Learning in Agent-Based Multiple Person Tracking_ TCSVT 2015

---

> ### Author Response · Authors · 2022-08-02
> **Response to Reviewer Tgjz**
>
> We thank the Reviewer for their valuable feedback and for recognizing our work as beneficial to the community to overcome the challenges of long-term occlusions. We also thank the Reviewer for highlighting other work before the deep learning era, which we will gladly add to the Related Work discussion.
>
> ### Reviewer comments that our method is complex, consisting of different components.
> We agree with the Reviewer that our overall method is rather complex, primarily due to difficulties associated with localizing object trajectories in BEV space based on a monocular video. Each step of our model, which had to be built and studied in isolation, provides several insights and discusses alternatives in the experimental section. As usual, based on lessons learned, we hope we can simplify and streamline methods for joint tracking and forecasting in the future.
>
> ### The Reviewer asks for an experiment comparing the projection approaches to represent objects in BEV presented in the MOTSynth experiment on the MOT dataset.
>
> In Fig. 4, we compare trajectory prediction for four different projection methods on the MOTSynth dataset: 3D ground truth positions projected on the scene plane (obtained from the GT depth), pixel positions transformed by the estimated homography, 3D positions reconstructed from image depth estimates and projected to BEV, and directly predicting motion in the pixel space.
>
> The comparison on real MOT data between the motion of the Kalman Filter in pixel space and BEV is demonstrated in the paper in Tab. 1. The results suggest a slight improvement of BEV motion in the global HOTA score of $54.08$ vs. $54.11$, but a significant increase of $7.08pp$ for long occlusions ($>2s$).
>
> We did not report results for the ground-truth projection and 3D depth points in the paper. In the following, we explain the absence of these experiments but share some insights with the Reviewer.
>
> In contrast to MOTSynth, we do not have ground-truth 3D positions for MOT sequences because the dataset only provides 2D object bounding boxes.
> Also, we experimented with 3D points directly extracted from the depth estimates on the MOT sequences. Unfortunately, temporally inconsistent depth values of pedestrians result in an average 3D localization error of $0.76m$ between consecutive frames. These extracted positions form very noisy trajectories, which are not accurate enough for trajectory forecasting and matching.
>
> Our experiences with these trajectories showed significantly inferior tracking performance than with our BEV reconstruction using homography why we did not proceed with these experiments.
>
> ### Reviewer reports broken hyper-references in the result tables.
> Thanks for pointing this out; we will fix this.
>
> ### The Reviewer asks for reporting of MOTA scores next to HOTA.
> We provide MOTA scores for the ablation study and final submission in Tab. 3, 4 (main), and Tab. 1, 2 in Supplementary.
>
> As per the Reviewer's request, we also report them here:
>
> | MOT17 (static) | |  | |  |   |  |  | MOT 20  |  |
> |-|----|---|--|-|-|-|-|-|-|
> | BYTE | CenterTrack   | CSTrack | FairMOT| JDE  | TraDeS  | TransTrack | QDTrack | BYTE   | CenterTrack  |
> 80.09  (+0.01) | 70.77  (+0.39) | 71.31  (+0.05) | 71.82  (+0.05) | 59.57  (+0.06) | 70.93  (+0.09) | 69.5  (+0.01) | 69.61  (+0.08) | 73.38  (+0.0) | 47.57  (+0.24) |
>
> MOTA has historically been the most prominent metric for multi-object tracking. However, the community has identified some drawbacks of the MOTA metric, and the public benchmarks like MOTChallenge and KITTI are slowly moving toward the HOTA evaluation metric. The main drawback for our studies is that MOTA dominantly focuses on detection performance and only little measures associations. As we do not modify the detected bounding boxes and only change their associations, the metric only marginally changes with improving associations.
>
> ### Reviewer asks for clarification on how we compute the egomotion for moving cameras.
>
> In the case of a moving camera, for each frame, we estimate a homography matrix $H_t$ and the optical flow of the image $O_t$. Given the homography $H_t$ we transform the pixel positions $x_t$ the corresponding BEV coordinates $X_t = H_t \cdot x_t$. We also use the optical flow to translate the pixels to $x^\prime_t = x_t + O_t$. Finally, we compute the translated points given the homography of the current timestep $X_t^\prime = H_t \cdot x_t^\prime$.
>
> We do this for all pixels of the semantic ground masks of two consecutive images. Hence we obtain two sets $\{ X_t \}$ and $\{X_t^\prime\}$. Now we compute the translation $t_t$ for the timestep $t$ that optimizes the $L2$ error between the two point sets (we do not consider rotations). The translation corresponds to the egomotion of the camera. Then, the egomotion $t_t$ is added to $X_t$ to construct absolute positions in BEV.
>
> We provide this explanation in the supplementary material C.2 -- we will clarify this in the revised paper.

---

> > ### Comment · Reviewer_Tgjz · 2022-08-08
> > **Response to authors**
> >
> > I appreciate the feedback from the authors and I have no further concerns. Look forward to the revised paper and released code on this work!

---

### Official Review · Reviewer_XrjC · 2022-07-11

**Rating:** 7
**Confidence:** 4
**Soundness:** 3 good
**Presentation:** 3 good
**Contribution:** 3 good

**Summary:**

This paper investigates the problem of long-term multi-object tracking. This is a relatively under-explored problem that worths investigation since most existing tracking methods focus on short-term tracklets (usually shorter than 2s). The main contribution of this paper is that they project the scene into bird-eye views (BEV) using homography transformation, and then apply trajectory forecasting and tracking jointly in the BEV space. The original version described in the main text supports only videos captured from fixed views, and an extending version described in the supplementary material steps further to support videos captured from moving cameras. Results on standard test sets suggest the proposed method improves tracking accuracy when combined existing trackers, and refreshes state-of-the-art when combined with the best performant tracker ByteTrack.

**Questions:**

1. As mentioned in the paper, forecasting multiple possible trajectories for lost tracklets bring more computation cost when perform association. It would be good to see how this affect the tracking accuracy/efficiency and how to do the trade-off.

2.  When dealing with moving cameras, homography is estimated for each single frame. This seems contradict conclusions in Figure 4(a), where it is shown that per-frame depth estimation is not stable so that the esitmated homography is largely affected. Is it possible to stablize the  depth estimation along the temporal dimension?

Typo:
L#309 0.43% -> 43.0%

**Limitations:**

The authors carefully discussed the limitation and and potential negative societal impact of their work in the main text.

**Strengths And Weaknesses:**

Originality:

Good. Existing MOT datasets and evluation metics focus more on short-time tracking accuracy. While many methods already perform fairly well on these metrics, they always fail when targets lost for relatively long time.  This paper introduces trajectory forecasting in the BEV space to handle long-term lost tracklets and shows promising results. To my knowledge, this is not well-explored in previous literature so the originality is good.

Quality:

Good. The entire pipeline is simple, easy to understand, and seems work well for long-term tracklets.
My biggest concern is about the forecasting module. From Table 1 it can be seen that applying Kalman Filter in the BEV space already performs well in terms of both prediction and tracking, and using advanced learning based methods seems do not bring too much gain. This is somehow below my expectation since Kalman Filter is such a simple linear model. Perhaps the problem lies in that most motion in the considered dataset is linear, and in this case, comparing different forecasting modules in a dataset with more complicated (non-linear) motion partterns may be helpful to validate the effectiveness of GAN.

Clarity:

Good but can be improved. In general, the presentation is clear and easy to follow. However, it would be better if key experimental conclusions are highlighted more clearly, especially for Table 1.

Significance:

Good. The long-term tracking problem is absolutely an imporant problem that is not well-explored due to limitation of existing datasets and metrics. This paper sheds new light on a promising direction towards solving this problem:  combine forecasting and tracking in BEV space.

---

> ### Author Response · Authors · 2022-08-02
> **Response to Reviewer XrjC**
>
> We thank the Reviewer for their valuable and positive feedback on our paper. We are thrilled that the Reviewer rates our paper to be accepted at the NeurIPS conference.
>
> ### Reviewer asks for the trade-off between accuracy and efficiency.
>
> We set the number of trajectory forecasts based on the performance/ accuracy of the validation set.
> The computational cost for predicting forecasts and matching to new detections increases linearly with the number of trajectory forecasts. However, the prediction and matching are only small parts of the computation of the tracker. In practice, we only predict $3-5$ trajectory forecasts for each lost object which does not significantly affect speed performance.
>
> The paper primarily focuses on building an entire pipeline from video to tracks studying different forecasting paradigms. Once the community starts to appreciate the benefit of BEV and trajectory forecasting, future work can focus on improving the end-to-end integration and efficiency of the algorithms.
>
> ### Reviewer asks about the improvement of depth consistency with temporal depth estimator.
>
> Having temporally stabilized depth estimates will most likely improve the robustness of the moving camera setup and the 3D point cloud for static scenes. Future work can try to use models such as [1].
>
> To explain our good results on moving cameras against the bad performance of 3D reconstruction of pedestrians in Fig. 4(a), we measured the temporal inconsistency of the projected 3D points.
>
> The magnitude of fluctuation is not the same for all pixels. The localization uncertainty of 3D points of smaller objects (like pedestrians) orthogonal to the scene plane is on avg. 6.5 times larger than inconsistencies on the ground plane.
> The experiment in Fig. 4(a) supports this finding as the frame-wise positions obtained from depth estimates are not very performant for trajectory forecasting.
>
> While not perfectly consistent, we can use the homographies estimated for different timesteps to compute the egomotion and BEV of the moving scenes. Noise in the real-world positions is averaged in the homography estimation because we use all pixels of the ground segmentation mask (usually a larger part of the image), and small fluctuations in the depth values are averaged in the homography estimation.
>
> [1]: Tananaev et al.: Temporally Consistent Depth Estimation in Videos with Recurrent Architectures

---

> > ### Comment · Reviewer_XrjC · 2022-08-10
> > **Post rebuttal comments**
> >
> > Thanks for the rebuttal！Most of my concerns are adequately addressed. I keep my postive rating.

---

> ### Author Response · Authors · 2022-08-08
> **Clarification of the Reviewer's questions**
>
> Dear Reviewer, we hope our previous comment clarified your main concerns. We are open to further discussion for the remaining time.

---

### Official Review · Reviewer_yuJE · 2022-07-12

**Rating:** 5
**Confidence:** 5
**Soundness:** 3 good
**Presentation:** 3 good
**Contribution:** 2 fair

**Summary:**

The paper investigates the exploitability of trajectory forecasting in multi target tracking. First, unmatched trajectories and unmatched detections are projected in bird-eye view (BEV), trajectories are then extended according to some model and tested for association. Projection in BEV also introduces visibility constraints that further reduces the search space of true matchings. The author validate their projection method against the true 3D position of pedestrians (in MOTSynth), then study the applicability and benefits of different trajectory forecasting models. They show that the proper combination can improve the HOTA score and reduce ID switches of 7 SOTA methods on MOT17 validation set. In the supplementary, the authors detail how their projection method can be extended to moving sequences.

**Questions:**

- Can the authors add the ID recall from [A] to the ablation studies in Tab.1 (the MOTChallenge evaluation kit does provide the means to compute it as it is required to compute the final ID F1) or, in alternative, justify why they don't find the metric appropriate and instead use AssA, AssPR and IDS.
- Can the authors help the reader putting in perspective the impact of the different contributions, i.e. working in bird eye view vs trajectory forecasting?
- Can the authors motivate the choice of 2m as a threshold for BEV matching and try to justify the difference between pixels and pixels 2D in Fig.4b?
- Can the authors help the reader understanding the end of line 322 "tracking results suggest otherwise"? How do the authors explain this unexpected result?

**Limitations:**

nothing else to add

**Strengths And Weaknesses:**

- (+) paper is well written and nicely structured, the reading is fluent; figures and tables are also helpful in the understanding of the key steps
- (+) the analysis of the contributions is carried out with an appropriate level of rigour
- (+) results are in favor of the proposed approach

- (-) HOTA is a complex and composed measure that tries to unify/project many aspects of tracking to a single scalar, so the authors have to compensate by also using AssA, AssPR as well as ID switches. Why did the authors thought these metrics to be more appropriate than ID recall [A] which measures the ability of a tracker to associate the same identity to a trajectory despite occlusions and interruptions? Isn't this exactly what the authors are trying to improve? Is the "ID Recall" in Fig.1 the one from [A]?

[A] Performance measures and a data set for multi-target, multi-camera tracking

- (-) From what I can see from Tab.1 most of the gain comes from a linear prediction method in 3D space. Other than that there is only a 0.5% HOTA left to gain from other trajectory forecasting methods. In this perspective I think the paper is overstating the importance of trajectory forecasting methods in multi target tracking and not helping the reader draw the correct conclusions. 3D tracking has been around since MOTChallenge 15 and kalman filtering even earlier and this seems to be the thing that explains 90% of the performance improvement (54.11-50.71)/(54.52-50.71).

- (-) For the same reason as above, I would have liked to see a comparison on 3D MOT 2015

- (-) In Fig.4 B we see a small difference between H and pixels L2, and a large difference between pixels L2 and pixels. The only way the reviewer can explain the difference between pixels L2 and pixels is through the use of a threshold in different domain, which seems to indicate that the threshold in BEV is less tight than the threshold in pixel space. As a matter of fact 2m seems like a very large margin to associate a trajectory and a detection, how did the authors choose this threshold? I think to remember that for both MOTA and IDF1 the threshold for 3D tracking was 1m instead of 2m (can be checked in the MOTChallenge evaluation kit).

- (-) lines 127-129 "simply applying trajectory prediction to MOT is not trivially possible" are misleading, the community has been doing this for years and it's also reported in Tab.1 Kalman Filter (pixel) with decent results.

---

> ### Author Response · Authors · 2022-08-02
> **Response to Reviewer yuJE Part1**
>
> We are happy that the Reviewer finds our paper well written and nicely structured, the analysis of our contributions to be rigorous, and the results to favor the proposed approach. We are grateful for the Reviewer's valuable feedback and are happy to respond to the Reviewer's questions.
>
> ### Reviewer asks to report additional tracking metrics.
>
> Per the Reviewer's request, we report  IDR metrics to Tab. 2 (below).
>
> |     | MOT17 (static) |                |                |                |                |                |                |                | MOT 20         |                |
> |-----|----------------|----------------|----------------|----------------|----------------|----------------|----------------|----------------|----------------|----------------|
> |     | BYTE           | CenterTrack    | CSTrack        | FairMOT        | JDE            | TraDeS         | TransTrack     | QDTrack        | BYTE           | CenterTrack    |
> | IDR | 78.61  (+0.39) | 65.25  (+6.25) | 67.53  (+0.87) | 66.23  (+0.53) | 56.08  (+1.09) | 67.12  (+1.06) | 61.39  (+0.01) | 62.17  (+0.68) | 66.44  (+0.34) | 35.87  (+3.23) |
>
> Next to the results presented in the paper, we can see that our tracking-by-forecasting method significantly improves the performance of many baseline trackers (e.g. $+6.25$ IDR for CenterTrack).
>
> ### Reviewer asks to explain the choice of HOTA (AssRe) metric over IDF1 (IDR).
>
> We decided to focus on the HOTA metric because it is the current metric that best balances both aspects of tracking task, detection, and association, especially compared to MOTA (measures mostly detection) and IDF1 (measures mainly association). Moreover, HOTA also allows us to perform a fine-grained analysis of the association performance by inspecting both association recall (AssRe) and precision (AssPr).
>
> ### Reviewer asks for clarification and definition of ID-Recall metric presented in Fig. 1.
>
> The ID-Recall metric (Fig. 1, defined in Suppl. F)  differs from IDR, defined in [A]. As IDR depends on factors such as trajectory length, the occlusion window, and global optimization behavior, we construct a straightforward but effective metric called ID-Recall. We only focus on occluded tracks and compute the ratio between successfully re-identified after the occlusion divided by the total number of occlusion cases for different occlusion lengths. We thank the Reviewer for pointing out that this should be clarified in the paper.
>
> ### Reviewer asks to report results on MOTChallenge 3D MOT'15 challenge.
> Unfortunately, we were unable to accommodate this request -- we were informed by MOTChallenge support team that submissions to this challenge are no longer allowed due to inaccuracies in the computation of the 3D groundtruth.
>
> ### Reviewer asks if it is possible to show the impact of the contribution of Bird Eye View Reconstruction and Trajectory Forecasting independently.
>
> Historically tracking methods have used simple linear models, eg., Kalman filter, to model motion in image (pixel) space.
> In contrast, forecasting methods operate in a metric, BEV space.
>
> As the Reviewer correctly observed, estimating a homography for a BEV transformation already results in a considerable performance boost of the linear model compared to the image-based counterpart (BEV linear > image linear), reducing IDSW in long-term occlusions by $16.1\\%$. However, as seen in Tab. 1, we can further improve performance ($17.4\\%$) using multimodal trajectory prediction models. This is not the case when operating in pixel space ($8.99\\%$).
>
> The analysis of different forecasting methods is part of the contribution of our paper and is only possible thanks to the BEV reconstruction. Trajectory forecasting models trained in pixel space (without a geometric meaning) would forfeit their purpose.

---

> ### Author Response · Authors · 2022-08-02
> **Response to Reviewer yuJE Part2**
>
> ### Reviewer asks about 2m threshold in the experiment shown in Fig. 4.
> For the experiment in Fig. 4, we empirically set the threshold to $2m$.
>
> Other choices for this threshold consistently led to the same conclusion: BEV consistently outperforms pixel and pixel (L2) motion. We will include an analysis of the sensitivity of this threshold in the final paper.
>
> ### Reviewer asks for an explanation of the difference between the motion models name pixels and pixels 2D in Fig. 4b.
>
> Both models use a linear motion model applied to pixel positions in the plot. However, the matching for the pixel model only has IoU matching while the final position of pixel 2D is transformed to BEV and has an additional L2 matching of $2m$ distance.
> Therefore, we find the advantage of transforming the prediction into a "normalized" metric BEV compared to only pixel space.
>
> Interestingly, the pixel 2D model performs worse for longer occlusions than the H model, which directly forecasts and matches in BEV. This experiment suggests the importance of forecasting in metric space and counteracting the non-linearity of the camera projection.
>
>
> ### The Reviewer ask for clarification line 322:"tracking results suggest otherwise"?
>
> In line 322, we discuss the single-generator GAN model, evaluated using $k = 20$ generated samples, as is the standard in the forecasting community.
>
> In this configuration, we follow the standard evaluation protocol applied in trajectory forecasting and report optimal (lowest) ADE and FDE.
>
> With the sentence, "tracking results suggest otherwise," we refer to the tracking performance of the model above, which yields the lowest overall HOTA score among all evaluated motion models.
>
> While successfully reducing the number of lost tracks for short ($-18.03 \\%$) and long ($-15.63 \\%$) occlusions, this configuration yields the lowest association precision.
>
> As the forecasting model produces $20$ samples for each lost track, several new detections are incorrectly associated to existing tracks, resulting in identity transfer errors. This result suggests a dis-alignment of evaluation metrics used in forecasting and tracking -- a better forecaster in terms of ADE/FDE does not necessarily lead to a better tracker.
> This is a known drawback of ADE/FDE metrics, which essentially measure only recall and not the precision of the forecasting output.

---

> ### Author Response · Authors · 2022-08-08
> **Clarification of the Reviewer's questions**
>
> Dear Reviewer,
> we hope our previous comment clarified your main concerns. We are open to further discussion for the remaining time.

---

### Official Review · Reviewer_cbmW · 2022-07-12

**Rating:** 6
**Confidence:** 5
**Soundness:** 4 excellent
**Presentation:** 3 good
**Contribution:** 3 good

**Summary:**

This paper explores the long-term occlusion problem in multi-object tracking, and proposes to using trajectory forecasting methods to compensate the tracking losts. The forecasting module is conducted in bird-eye-view. The method can advance state-of-the-art trackers on the MOT Challenge dataset.

**Questions:**

See weaknesses.

**Limitations:**

The authors discussed the limitations in the paper.

**Strengths And Weaknesses:**

Overall, I like the idea in this paper. I agree that most current multi-object trackers do not tackle long-term occlusions. The solution in this paper is sound and makes sense to me. However, I feel that the work of this paper hasn’t been finished yet. It doesn’t have supportive experiments. I hope the authors can improve the draft with the following comments.

- Strengths
1. The paper is well-written and easy to follow.
2. The paper tackled an existing problem in multi-object tracking and the solution is reasonable.

- Weaknesses and Suggestions
1. It makes sense that long-term occlusion rarely occurs in MOT Challenge datasets. However, the paper should have supportive experimental results. Current results, boost tracker by ~0.1 HOTA cannot prove the effectiveness of the method. I suggest the authors try to test the oracle of forecasting matching and show the advanced percentage of the method in this part of errors.
2. How about doing experiment in autonomous driving datasets? The dataset has camera parameters.

---

> ### Author Response · Authors · 2022-08-02
> **Response to Reviewer cbmW Part1**
>
>
> We are happy that the Reviewer finds our paper well-written, our method sound, and reasonable. We thank the Reviewer for their feedback.
>
> ### The Reviewer is asking for additional experiments to support the strength of our paper and questions the significance of our methods because we can only improve the HOTA performance by ~0.1.
>
> Even though long-term occlusions present the biggest challenges to modern trackers, they are statistically rarer than most fully visible short tracks or shortly-occluded tracks.
>
> Thus, improving a smaller number of the most challenging cases has only a marginal impact on the overall performance.
>
> For this reason, in Fig. 1 and Tab. 1, we highlight results obtained when only evaluating object tracks that undergo occlusions. As can be seen, our method successfully resolves over $17.4\\%$ of long-term occlusions, which are not solved by prior work.
>
> We believe this increase is a significant and meaningful contribution to the community, in addition to our analysis of how current trends in trajectory prediction affect tracking performance.
>
> ### The Reviewer suggests expanding experimental evaluation using an autonomous driving dataset.
> Per the Reviewer's recommendation, we analyzed one of the largest vision-based datasets for autonomous driving, BDD100K (https://www.bdd100k.com/), and widely-used KITTI (http://www.cvlibs.net/datasets/kitti/). Our analysis shows that less than $0.6\\%$ tracks in BDD100K and $4\\%$ tracks in KITTI contain occlusion gaps longer than $2s$. In contrast, in MOTChallenge, $19.4\\%$ contain long (over $2s$) occlusion gaps. This is likely due to the semi-automated annotations process in these large-scale datasets, where objects reappearing after occlusions are often assigned new identities.
>
> Therefore, autonomous driving datasets are at the moment, not well suited for studying long-term tracking.
>
> In contrast, identity preservation is essential for video editing, safety camera analysis, or social robots interacting with humans. In these scenarios, we often only have access to a single RGB camera without additional 3D sensor data. In these cases, BEV reconstruction combined with trajectory forecasting greatly contributes to improving long-term tracking.
>
> The Reviewer rates our soundness as excellent, presentation, and contribution as good; however, they recommend rejection. We would appreciate insights into the final rating.

---

> > ### Comment · Reviewer_cbmW · 2022-08-09
> > **Responses**
> >
> > I have read the responses from the reviewers, and they addressed my concerns. I will increase my rating after the Reviewer-Meta Reviewer Discussion phase.
> >
> > I recommend the authors highlight these performance analyses of occlusions in their abstract/introduction to show the appealing benefits of the work.

---

> ### Author Response · Authors · 2022-08-08
> **Clarification of the Reviewer's questions**
>
> Dear Reviewer,
> we hope our previous comment clarified your main concerns. We are open to further discussion for the remaining time.

---

### Meta-Review · Area_Chair_scGt · 2022-08-23

**Recommendation:** Accept
**Confidence:** Certain

**Metareview:**

The paper initially had mixed reviews 4567. The main concerns of the reviewers were:
1. can better show the improvement on long-term occlusions (cbmW)
2. lack of results on autonomous driving datasets w/ camera parameters. (cbmW)
3. Questions about the evaluation metrics used (yuJE, Tgjz)
4. In Tab 1, most of the HOTA gain comes from linear prediction in 3D space, i.e., Kalman filters.  (yuJE)
5. comparison on 3D MOT 2015 (yuJE)
6. missing ablation study on association threshold (yuJE)
7. what is the tracking / efficiency tradeoff for forecasting (XrjC)
8. how to deal with moving cameras (XrjC, Tgjz)
9. complex pipeline requires training separate sub-models (Tgjz)
10. ablation study on the different view projection methods (Tgjz)

The authors wrote a response to address these concerns. The reviewers were largely satisfied with the response. Reviewer yuJE still had a concern about the message of the paper (Point 4; Reviewer's point [A.1]), and responded:
> The authors replied by assessing that working in BED is already trajectory forecasting. I do not agree with that, that is just 3D or metric tracking. And metric tracking + kalman filter, which explain 90% of the contribution of the paper, should not be advertised as novelty, nor as trajectory forecasting. This view that I am suggesting here, clearly help the reader in understanding that trajectory forecasting is really of little help in MTT (~0.5% HOTA), which is the opposite of what the paper is claiming.

> As I see it, the paper has merits, e.g. ways to go from image to BED in static as well as in moving sequences, but that is not the story told by this paper (the most interesting part being in the supplementary material).

Nonetheless, the final ratings were positive (5667), and the reviewers appreciated the problem solution to handle long-term occlusions, and brings a promising direction for future research.  The AC agrees and recommends accept. The authors should revise the paper according to the reviewers' comments and the discussion.


**Award:**

No

---

### Decision · Program_Chairs · 2022-09-14

Accept